# Low-noise frequency-agile photonic integrated lasers for coherent ranging

Grigory Lihachev[1,4], Johann Riemensberger [1,4], Wenle Weng [1,3,4], Junqiu Liu [1], Hao Tian [2], Anat Siddharth [1], Viacheslav Snigirev[1], Vladimir Shadymov[1], Andrey Voloshin[1], Rui Ning Wang [1], Jijun He[1], Sunil A. Bhave [2] & Tobias J. Kippenberg [1✉]

Frequency modulated continuous wave laser ranging (FMCW LiDAR) enables distance mapping with simultaneous position and velocity information, is immune to stray light, can achieve long range, operate in the eye-safe region of 1550 nm and achieve high sensitivity. Despite its advantages, it is compounded by the simultaneous requirement of both narrow linewidth low noise lasers that can be precisely chirped. While integrated silicon-based lasers, compatible with wafer scale manufacturing in large volumes at low cost, have experienced major advances and are now employed on a commercial scale in data centers, and impressive progress has led to integrated lasers with (ultra) narrow sub-100 Hz-level intrinsic linewidth based on optical feedback from photonic circuits, these lasers presently lack fast nonthermal tuning, i.e. frequency agility as required for coherent ranging. Here, we demonstrate a hybrid photonic integrated laser that exhibits very narrow intrinsic linewidth of 25 Hz while offering linear, hysteresis-free, and mode-hop-free-tuning beyond 1 GHz with up to megahertz actuation bandwidth constituting $1.6 \times 10^{15}$ Hz/s tuning speed. Our approach uses foundry-based technologies - ultralow-loss (1 dB/m) $Si_3N_4$ photonic microresonators, combined with aluminium nitride (AlN) or lead zirconium titanate (PZT) microelectromechanical systems (MEMS) based stress-optic actuation. Electrically driven low-phase-noise lasing is attained by self-injection locking of an Indium Phosphide (InP) laser chip and only limited by fundamental thermo-refractive noise at mid-range offsets. By utilizing difference-drive and apodization of the photonic chip to suppress mechanical vibrations of the chip, a flat actuation response up to 10 MHz is achieved. We leverage this capability to demonstrate a compact coherent LiDAR engine that can generate up to 800 kHz FMCW triangular optical chirp signals, requiring neither any active linearization nor predistortion compensation, and perform a 10 m optical ranging experiment, with a resolution of 12.5 cm. Our results constitute a photonic integrated laser system for scenarios where high compactness, fast frequency actuation, and high spectral purity are required.

[1] Institute of Physics, Swiss Federal Institute of Technology Lausanne (EPFL), CH-1015 Lausanne, Switzerland. [2] OxideMEMS Lab, Purdue University, West Lafayette, IN 47907, USA. [3] Present address: Institute for Photonics and Advanced Sensing (IPAS), and School of Physical Sciences, The University of Adelaide, Adelaide, South Australia 5005, Australia. [4] These authors contributed equally: Grigory Lihachev, Johann Riemensberger, Wenle Weng. ✉email: tobias.kippenberg@epfl.ch

Low-phase-noise lasers[1–4] are imperative in a wide range of technological and scientific applications, ranging from distributed fibre sensing[5], coherent LiDAR[6–10] to microwave photonics[11]. Over the past decade, the development of heterogeneously integrated lasers has led to a new class of CMOS-compatible highly integrated laser sources[12–14] that are now commercially employed in data-centre interconnects. The fundamental linewidth, i.e., the phase noise, of lasers is given by the modified Schawlow-Townes linewidth limit[15,16], which dictates that low-loss laser cavities with a high number of photons stored in the cavity allow inherently low phase noise. In addition to quantum noise, thermodynamical noise, such as thermo-refractive noise due to refractive index fluctuations, constitutes another limit[17,18]. To date, the lowest laser phase noise of compact semiconductor lasers is achieved by self-injection locking with discrete crystalline resonators (sub-Hz white frequency noise level)[19,20], and extensively to silicon-based lasers[21,22]. Using $Si_3N_4$ TriPleX waveguides[23,24] as first demonstrated by the pioneering work of the group of K.J. Boller, has enabled hybrid integrated lasers with Hz-level Lorentzian linewidth, that have shown steady improvements[1]. Using weak confinement $Si_3N_4$ waveguides and laser self-injection locking it has culminated recently in the demonstration of frequency noise of 0.006 $Hz^2$/Hz at 4 MHz offset[3]. A particularly interesting application for integrated photonics-based low noise lasers is coherent FMCW LiDAR. Recently, autonomous driving and areal mapping have increased the interest in such sources and a fully hybrid integrated low-noise, high and frequency-agile source could hence unlock further applications of coherent FMCW LiDAR. Laser phase noise limits the maximum operating distance and ranging precision in FMCW LiDAR[25,26]. However, a key requirement for FMCW at long range is in addition to low phase noise, frequency agility, i.e to achieve fast, linear and hysteresis-free tuning[26]. Currently, integrated laser sources require external linearization or pre-compensation. Most digital approaches towards photonic integrated FMCW LiDAR, which employ injection-locking of a high power laser diode to an electro-optically modulated sideband of a coherent laser can deliver excellent linearity and low noise[27,28], but today require bulk optical circulators and fibre laser oscillators for operation.

Here we demonstrate a laser that combines both low noise and frequency agility - enabling laser tuning at $1.6 \times 10^{15}$ hertz/second. While both have been attained separately in integrated devices, and faster tuning (e.g. MEMS VCSEL lasers) and even lower phase noise has been attained for individual lasers, our work marks these two properties are simultaneously achieved in an integrated device, while maintaining fast (MHz) tuning and narrow linewidth (low phase noise). This is attained by using self-injection locking (SIL) of an III-V InP laser to an ultralow-loss $Si_3N_4$ microresonator[2,29,30] monolithically integrated with AlN MEMS-based actuators[31,32], we achieve both frequency agility and narrow linewidth, exhibiting phase noise that is on par at mid range offsets with fibre lasers - the workhorse for fibre sensing. Owing to the high $Q$ of the $Si_3N_4$ microresonator resonances, the InP laser shows a reduced intrinsic linewidth of ~25 Hz. Using the AlN piezoelectrical actuators engineered based on novel contour mode cancellation and differential drive schemes allows the photonic microresonator to be frequency-modulated via the stress-optic effect with a flattened response up to the actuation frequency of 10 MHz - order of magnitude improvement[33] due to the planar co-integration. This enables a class of compact LiDAR sources that do not require external linearization of the FMCW signal. We generate narrow-linewidth triangularly chirped lasers capable of chirp repetition frequencies as high as 800 kHz and nonlinearities as low as 1% without digital predistortion or complex direct microwave signal synthesis and perform an FMCW LiDAR demonstration at 100 kHz chirp frequency. The versatility permitted by the optical and mechanical properties of the system shows great promise in applications including, field-deployable frequency referencing[34], frequency-agile rapid-scanning spectroscopy[35,36] and low-cost FMCW LiDAR engines[32].

## Results

**Hybrid self-injection-locked laser system**. As illustrated in Fig. 1a, the hybrid laser system comprises a III–V laser chip with a distributed feedback (DFB) structure and a photonic chip-based ultralow-loss $Si_3N_4$ resonator with a monolithically integrated AlN piezoelectrical actuator. The DFB laser diode is mounted on a 3D piezoelectrical translation stage and butt-coupled to the $Si_3N_4$ photonic chip as shown in Fig. 1b, c operating at a lasing wavelength of 1556 nm with a free space output power of up to 120 mW. The $Si_3N_4$ photonic chips are fabricated using the photonic Damascene reflow process[30,37,38], and feature intrinsic quality factor $Q_0 > 1.5 \times 10^7$. Frequency-dependent transmission, reflection and cavity linewidth data is presented in the SI Fig. 5 for all chips used in this work. Made from polycrystalline AlN as the main piezoelectric material, the actuator has molybdenum (Mo) and aluminium (Al) as the bottom (ground) and the top electrodes, respectively[31] as shown in Fig. 1d. Applying a voltage between the electrodes tunes the microresonator frequency via the stress-optic effect[39]. Such hybrid packaging approach[23] and also heterogeneous integration with InP laser and $Si_3N_4$ PIC fabricated on a single silicon substrate[21], have recently been demonstrated with $Si_3N_4$ microresonators. By tuning the current of the laser diode, we sweep the relative frequency between the laser and the resonator modes to attain self-injection locking via the coupling of counter-propagating microresonator modes induced by backscattering predominantly from the core-cladding interface[40] (cf. SI Figs. 5 and 1e). The gap between the laser chip and the $Si_3N_4$ photonic chip is adjusted for optimal feedback phase[41], which yields the maximum self-injection locking range of up to 2.1 GHz (cf. Fig. 1e).

Self-injection locking in this manner has been attained in previous work, however, the ultralow-loss photonic integrated resonator enables the substantial reduction in the phase noise via self-injection locking that is hitherto only surpassed in low confinement $Si_3N_4$ microresonators[2] without non-thermal actuation or crystalline microresonators[19]. In our work we only consider the linear regime of laser operation by adjusting the feedback phase and keeping the output optical power below 1.5 mW. In order to achieve frequency agility, we bias the diode in the centre of the locking plateau. In this manner, changes in the microresonator frequency will maintain injection locked operation and therefore lead to frequency tuning (cf. Fig. 1f). The AlN actuator will therefore transduce the applied voltage directly to changes in the optical frequency. Figure 1h shows the range over which the photonic resonator frequency can be tuned while still maintaining injection locking (corresponding to a tuning range of up to 2.1 GHz).

**Laser frequency noise measurements**. To measure and confirm the very low phase noise of the laser, we adopt two approaches (see Fig. 2a) to measure the frequency noise power spectral density $S_\nu(\Omega)$ (single-sided PSD, in units of $Hz^2$/Hz), including (1) beating the injection-locked laser with an ultra-narrow-linewidth reference laser (see the SI for details) and measuring the beat signal's frequency noise spectrum with an electric spectrum analyser (ESA), and (2) optical cross-correlation-based noise spectrum characterisation[42] using two auxiliary lasers. (cf. Figs. 2b and SI). Since the laser linewidth narrowing factor, i.e. the ratio of the free-running laser linewidth to the linewidth of the injection-locked laser, is quadratically

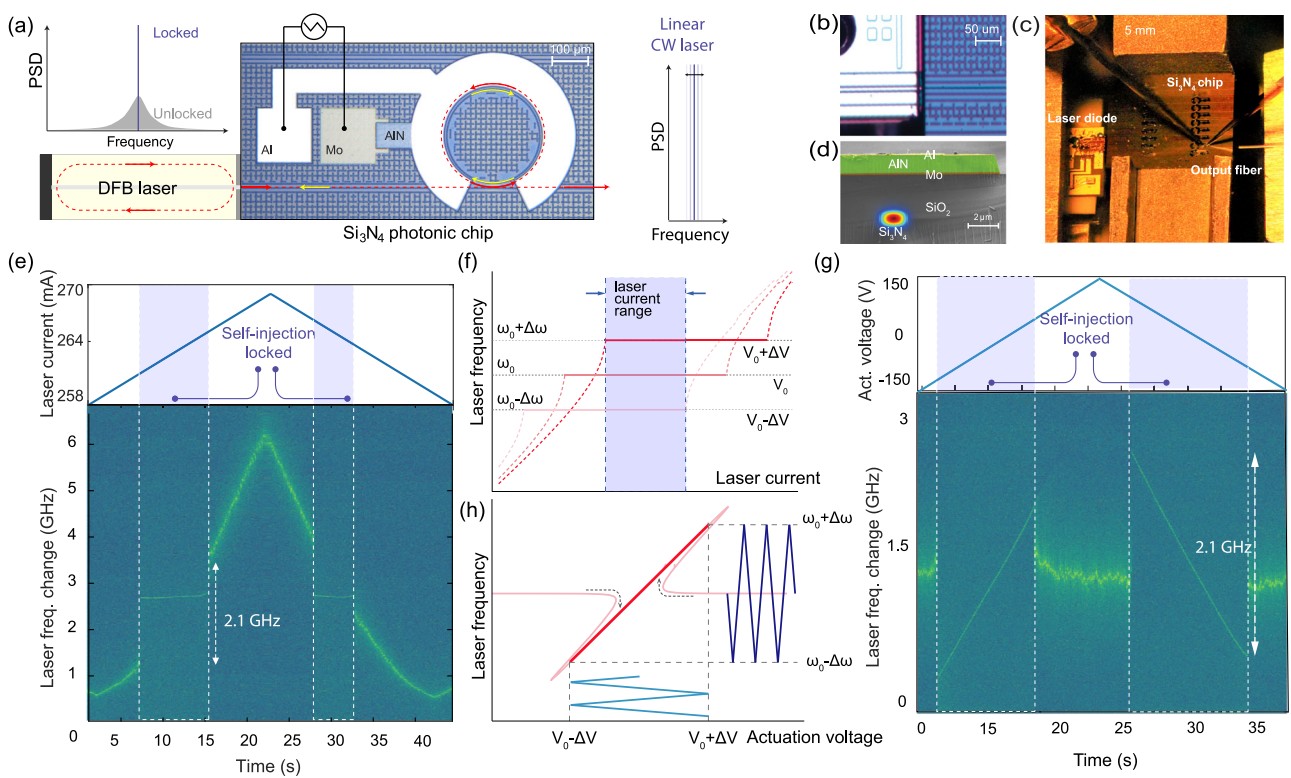

**Fig. 1 Schematic of the hybrid integrated laser system. a** Principle of laser linewidth narrowing via laser self-injection locking. The laser frequency tuning is realised by applying a sweeping electrical signal on the monolithically integrated AlN actuator. **b** Optical micrograph showing the DFB laser butt-coupled to the $Si_3N_4$ photonic chip. **c** Photo of the experimental setup with DFB laser (left, mounted on piezoelectric stage), $Si_3N_4$ chip (middle), output lensed fibre, probes for piezoactuator (top). **d** False-coloured scanning electron microscope (SEM) image of the sample cross-section, showing the piezoelectric actuator integrated on the $Si_3N_4$ photonic circuit. The piezoelectric actuator is composed of Al (yellow), AlN (green) and Mo (red) layers on top of $Si_3N_4$ buried in $SiO_2$ cladding. **e** Spectrogram showing laser frequency change upon the linear tuning of the diode current, measured for 190.7 GHz FSR microresonator, dashed areas correspond to the range where the laser is self-injection locked (featured with minimal lasing frequency fluctuations). **f** Schematic of the tuning of the laser frequency. Different voltage levels applied to the piezoactuator correspond to different microresonator resonance frequencies, thus leading to the different frequencies of the laser when the laser current is in the range for self-injection locking. **g** Spectrogram of laser frequency change upon the linear tuning of the cavity resonance by piezoelectric actuator, measured for 190.7 GHz FSR microresonator, dashed areas correspond to the range where the laser is self-injection locked to the shifting resonance. **h** Schematic of linear laser frequency tuning with integrated piezoactuator. By applying the triangular voltage ramp to the piezoactuator, we transduce the cavity resonance shift induced by the piezoactuator to the triangular laser frequency change while operating inside the locking range.

proportional to the $Q$ of the resonator mode to which the laser is self-injection-locked[41], the high loaded $Q$ of the $Si_3N_4$ microresonator (see Fig. 2c) can significantly reduce the optical linewidth and improve the side mode suppression ratio (SMSR). We present simulations of the self-injection locking dynamics in the SI. Figure 2d shows the optical spectrum of the laser while it is self-injection locked, demonstrating an SMSR of 60 dB. The relative intensity noise (RIN) of the laser is displayed in the SI. Figure 2e shows the frequency noise spectra of the hybrid lasers in free-running condition and self-injection locking states, respectively. For the injection locking, three microresonators with distinct sizes and different free spectral ranges (FSRs) of 190.7, 9.87 and 2.45 GHz are tested. In general, the self-injection locking suppresses the frequency noise of the laser by more than 30 dB across the entire spectrum. At frequencies below 1 kHz, technical noises due the ambient temperature fluctuations, and the coupling gap instabilities caused by acoustic vibrations (which can be eliminated by packaging) leads to hybrid integrated laser RIN and also to a transduction to frequency noise at offsets < 1 kHz (see SI). The optical crosscorrelation-based characterisation reveals that the laser frequency noise reaches a plateau (white noise floor) of only 8 $Hz^2$/Hz for the 2.45 GHz device, 10 $Hz^2$/Hz for the 9.87 GHz and 300 $Hz^2$/Hz for the 190.7 GHz device at 3 MHz offset. The frequency noise power spectral density was found

to be in good quantitative agreement with fundamental thermo-refractive noise (TRN) limit[43] of the $Si_3N_4$ microresonator at mid range offsets from 5 to 100 kHz. Thus, we demonstrate using high confinement $Si_3N_4$ platform the laser performance limited by thermo-refractive noise, which has only been shown in $Si_3N_4$ low-confinment waveguides[2,3]. We note that this intrinsic linewidth can be further reduced by using microresonators with even larger optical mode volume and therefore reduced TRN. To illustrate the performance of this integrated laser, we compare the laser frequency noise to two commonly used lasers. Figure 2f reveals that our hybrid integrated laser is better than a state of the art commercial external cavity diode laser (ECDL,Toptica CTL) and on par with a commercial fibre laser (NKT Koheras Adjustik E15) at the offset frequency range of 1–50 kHz. Another method to quantify and compare the linewidth from the measured frequency noise spectral density $S_\nu(f)$ is to invoke the beta-line[44]. By integrating frequency noise up to the frequency of the interception point with a line $S_\nu(f) = 8 \cdot \ln(2) f / \pi^2$, we obtain an area A which we use for FWHM measure of the linewidth $(8 \cdot \ln(2)A)^{1/2}$. The full width at half maximum linewidth, which is calculated by integration of the frequency noise from the beta-line to the inverse integration time, is for the 9.87 GHz FSR $Si_3N_4$ device 7.5 kHz at 1 ms integration time, 18.7 kHz at 10 ms, and 21.5 kHz at 100 ms. For the 190.7 GHz FSR

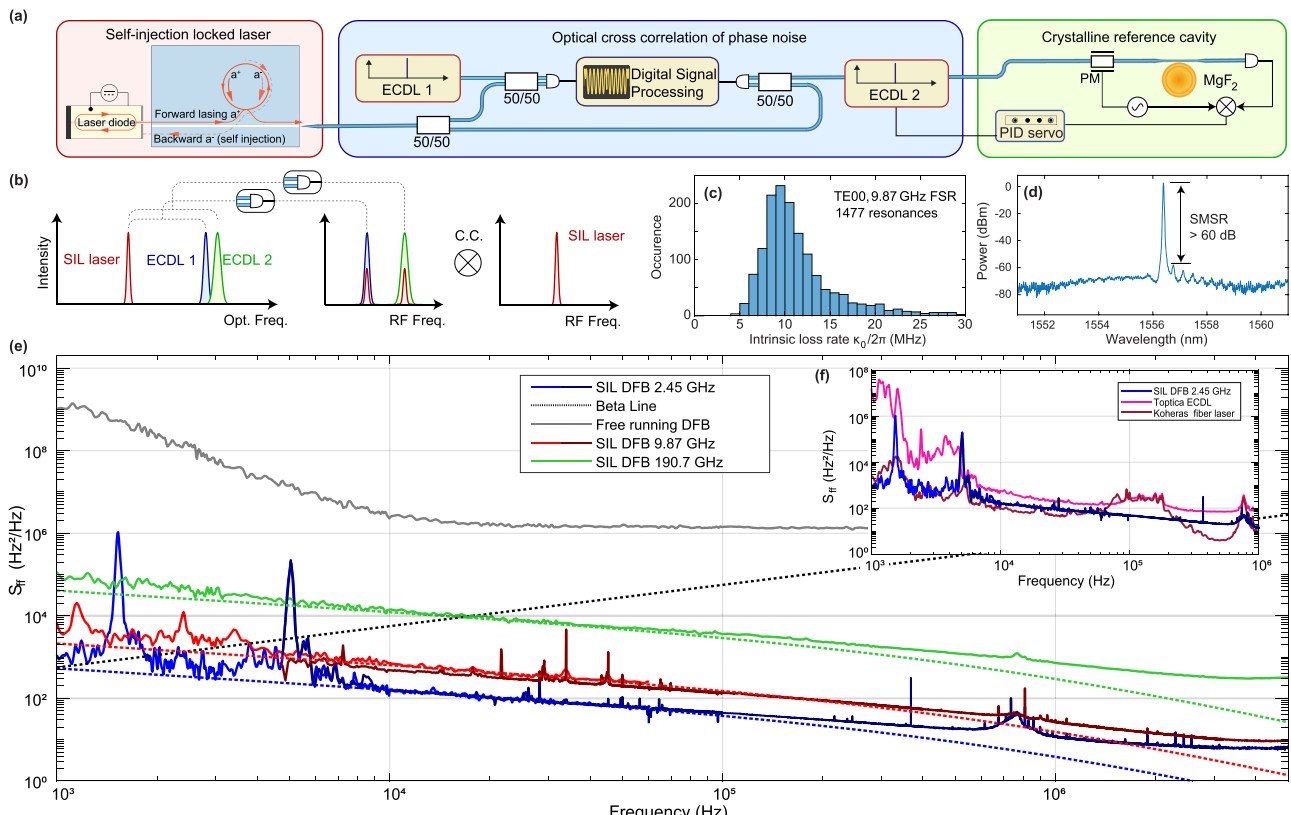

**Fig. 2 Spectral purity of the self-injection-locked laser. a** Experimental schemes of laser frequency noise measurements, using optical cross-correlation technique or heterodyne beat with the external cavity diode laser (ECDL) locked to a high-Q crystalline resonator using the Pound-Drever-Hall technique. **b** Schematic in the frequency domain of optical cross-correlation (C.C.) technique. **c** The $Si_3N_4$ photonic chips with an FSR of 9.87 GHz have an intrinsic loss of $\kappa_0/2\pi < 10$ MHz, corresponding to a quality factor of $Q_0 > 20 \times 10^6$. The $\kappa_0/2\pi$ histogram of 1477 $TE_{00}$ resonances from a 9.87 GHz FSR microresonator is shown. **d** Optical spectrum of the SIL DFB emission. **e** Single-sided PSD of frequency noise of the hybrid integrated laser system upon self-injection locking to microresonators with FSRs: 190.7 GHz (green), 9.87 GHz (red), 2.45 GHz (blue) and free running regime (grey). Dark colour traces correspond to optical cross-correlation data, light colour traces to the heterodyne beat method. The dotted red, green, blue lines indicate the calculated thermo-refractive noise limit for $Si_3N_4$ microresonators with different FSR. **f** The inset shows a comparison of frequency noise of SIL DFB with a commercial ECDL (Toptica CTL) and a commercial fibre laser (NKT Koheras Adjustik E15).

$Si_3N_4$ device the integrated linewidth is 43.27 kHz at 1 ms integration time, 73.7 kHz at 10 ms and 81.7 kHz at 100 ms. For the 2.45 GHz FSR $Si_3N_4$ device the integrated linewidth is 14 kHz at 1 ms integration time, 127 kHz at 10 ms and 130 kHz at 100 ms (cf. SI Fig. 6).

**Frequency-agile tuning**. We next demonstrate the frequency agility of our hybrid laser. To this end, we carry out frequency-modulation of the self-injection-locked laser by applying a time-varying voltage to the single integrated AlN actuator (cf. Fig. 3a) manufactured on top of the 190.7 GHz ring resonator on a square (4.96 × 4.96 mm) non-apodized chip. Triangular ramp signals of ramping frequencies from 10 kHz to 800 kHz are generated with an arbitrary frequency generator and amplified to 150 $V_{PP}$ which induces a 1.1 GHz laser frequency modulation. The tuning efficiency is composed of two contributions: first, the microscopic photoelastic effect and, second, the change of ring radius due to the generated mechanical stress. The latter leads to a 30% increase in tuning efficiency for the 190.7 GHz resonator compared to larger ring resonators. In the self-injection locking range, the change of microresonator frequency imprints directly on the laser output frequency, even without additional feedback on the pump current of the laser. The time-varying laser output frequency is characterised by measuring a heterodyne beatnote with a reference ECDL (free running Toptica CTL) on a fast photodetector. We define chirp nonlinearity as the root mean square (RMS)

deviation of the measured frequency tuning curve from a perfect triangular ramp that is determined with least-squares fitting. The phase noise PSD of the tuned laser can also be directly retrieved by Hilbert's transform from the heterodyne beat note (cf. SI Fig. 10). Figure 3b, c summarise the main results of the heterodyne beat experiment with the SIL laser locked to a 190.7 GHz microresonator. The large tuning range of >1 GHz at high ramping speeds up to 800 kHz, with small chirping RMS non-linearities below 1% as shown in Fig. 3b showcases the remarkable frequency agility of our system. This excellent linearity and the almost vanishing hysteresis of the monolithically integrated AlN actuator[32] facilitates the generation of highly linear triangular chirps for modulation frequencies up to 100 kHz without the need for active or passive linearization. Figure 3d presents the processed laser frequency spectrograms and the corresponding nonlinearities at five different ramping frequencies respectively. At 10 kHz modulation frequency, the achieved RMS nonlinearity is as low as 600 kHz (relative nonlinearity $5 \times 10^{-4}$), which only degrades slightly for 100 kHz tuning rate to 1.5 MHz. In Fig. 3c, we plot the frequency-dependent transduction from the frequency modulation amplitude of the first 17 harmonics of each modulation frequency (10 kHz–800 kHz) from the experimental data presented in Fig. 3d. The peak around 900 kHz matches well with the first mechanical mode of the chip presented in Fig. 3e. The 4 MHz low pass cut-off of the high voltage amplifier is indicated in red. Together with more than 1 GHz tuning range,

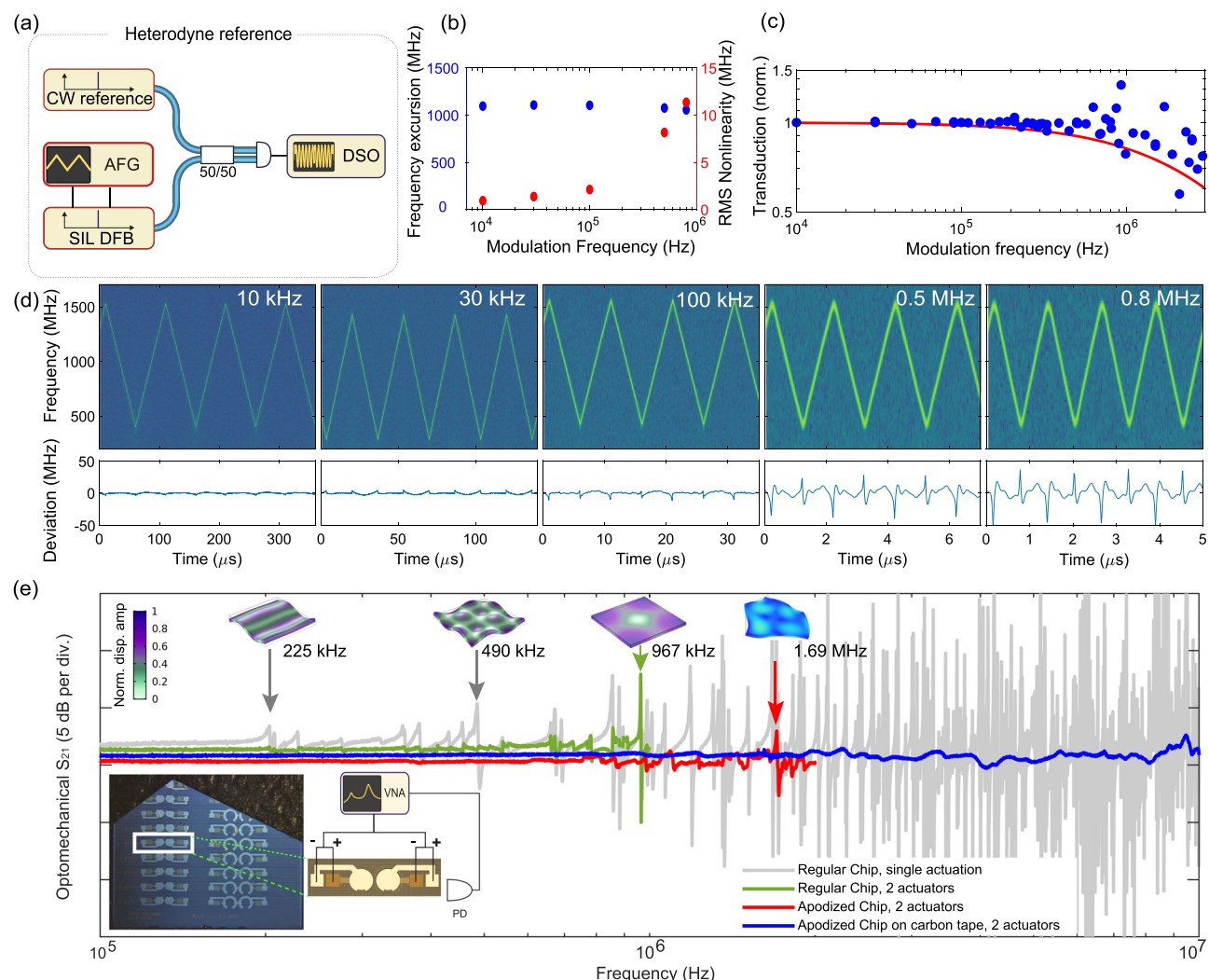

**Fig. 3 Frequency-agile tuning with integrated AlN piezoactuator. a** Experimental setup for heterodyne beat note characterisation of the frequency-agile hybrid-integrated laser. A continuous-wave (CW) external-cavity diode laser is used as a reference, and the beatnote is recorded on a fast oscilloscope (DSO) and analysed with short-time Fourier transforms. **b** Frequency excursion (blue) and residual root-mean-square (RMS) nonlinearity (red) of triangular laser chirps. The AlN actuator is driven with a peak-to-peak amplitude of 150 V. **c** Piezo-voltage to laser frequency transduction is calculated from the harmonic spectral content of the laser frequency chirp and the best fitted perfect triangular frequency chirp (red). **d** Time-frequency spectrogram of the heterodyne beat-notes for different triangular chirp repetition frequencies. Bottom row: residual of least-squares fitting of the time-frequency traces with symmetric triangular chirp pattern. **e** Suppression of photonic chip mechanical resonances. Measured responses of the stress-optic actuation for 190.7 GHz FSR microresonator using disk-shaped piezoactuator with single actuation (grey), dual actuators with difference-actuation for a square $Si_3N_4$ chip (green), an apodized chip (red) and an apodized chip on a carbon tape (blue). Insets: three mechanical modes of regular $Si_3N_4$ chip (225 kHz, 490 kHz, 967 kHz) and apodized chip (1.69 MHz) simulated with FEM, eigenfrequencies denoted by the arrows with displacement amplitude profile visualisation. Lower left: photo of the apodized chip with the dual-actuator configuration, exact chip dimensions are provided in the SI. Experimental schematic for difference driving of dual actuator.

such actuation bandwidth exceeds the performance of common benchtop laser systems that rely on bulk piezo making this laser source an ideal candidate for direct implementation in long-range FMCW LiDAR systems[45], that can operate at rates reaching megapixel-per-second.

**Flattening actuation response via photonic chip mechanical modes suppression.** To achieve the optimal performance of the hybrid integrated laser as an FMCW LiDAR engine at high measurement rates, in addition to a large optical frequency excursion B (which determines LiDAR resolution $c/2B$), a flattened actuation transfer function is highly desired for minimising chirping nonlinearity. The lower inset of Fig. 3e shows the setup of the actuation response measurement. In this measurement, the

actuation voltage derived from a vector network analyser (VNA) is applied on the actuator, and a laser is frequency-tuned to sit on the side of resonance, to measure the response. Figure 3e presents the measured optomechanical response of the single-actuator configuration (grey) and the dual-actuator configuration (green, red and blue). As shown in Fig. 3e (grey), the fabricated AlN piezoelectrical monolithic actuator excites many mechanical bulk or contour modes of the photonic chip, leading to a nonflat actuation response. The inset of Fig. 3e shows finite element simulations of the flexural modes of the photonic chip, that match the observed actuation resonances. The increasing mode density of the $Si_3N_4$ photonic chip with actuation frequency severely limits the flat effective actuation bandwidth. We mitigate this effect, first, by developing the active cancellation scheme with a difference-actuation. In this scheme, an additional AlN actuator

with the same geometry is fabricated adjacent to the micro-resonator. The two actuators are driven by the same signal but with a 180-degree phase shift to cancel the actuation of the mechanical modes of the photonic chip. As a result, while the stress-optic effect exerted on the microresonator is the same, the excitation of mechanical resonances can be effectively suppressed as shown in Fig. 3e (green). The scheme effectively reduces modes below 1 MHz, mainly cancelling the flexural modes due to far-field destructive interference, a scheme inspired by nanomechanical membranes[46]. Many mechanical modes of relatively low resonance frequencies are flexural modes whose vibrations are caused by transverse standing waves. The bulk mechanical modes whose vibrations are caused by longitudinal standing waves (displacement amplitude profile visualisation shown at 967 kHz frequency) can be eliminated by judiciously shaping the geometry of the photonic chips[47]. To improve the actuation further, and suppress the bulk acoustic modes, we next apodized the photonic $Si_3N_4$ chip by dicing the released chip. We observe a reduction in the number of bulk mechanical modes in an apodized photonic chip. Figure 3e (red) shows that the mechanical resonances below 1 MHz are completely removed with the first resonance of apodized chip at 1.69 MHz, matching the finite element simulation (FEM). We further flatten the actuation response by attaching the apodized chip on a piece of carbon tape and then differentially driving the actuators as explained before. In this way, both the flexural and the bulk mechanical modes are damped up to the first fundamental high-overtone bulk acoustic resonator mode (HBAR) at 17 MHz[32]. The active cancellation scheme on an apodized chip placed over a carbon tape limits the variations of the actuation response within 1 dB yielding a record-flat response bandwidth of nearly 20 MHz (see the SI for the full plot). Such flat actuation response can improve the linear chirping performance of the FMCW LiDAR (see the SI for the analysis of the required actuation bandwidth), and in particular allows an increase of the FMCW signal frequency, thereby directly increase the speed of the LiDAR to beyond megapixel-per-second rates.

**Optical FMCW LiDAR using the hybrid integrated laser**. As an actual demonstration of the potential of the hybrid integrated laser, we perform optical FMCW LiDAR mapping in the laboratory environment. Importantly, we can – due to the excellent linearity, low hysteresis, and narrow linewidth – perform ranging without any adaptive clock sampling and without pre-distortion linearization. Figure 4a shows the experimental setup of FMCW LiDAR measurement, for the description of the experiment refer to Methods. We apply a triangular chirp with 100 kHz frequency to piezoelectric AlN actuator to obtain 1.2 GHz optical frequency excursion of self-injection locked laser, corresponding to 12.5 cm resolution in distance measurement. Beam steering is realised using mechanical galvo scanner with two mirrors. For the ranging target scene, we use a polystyrene foam donut in front of a PC monitor 10 m away from the laser collimator (Fig. 4b). We record a beat signal of light reflected from the target and the laser in the local oscillator path on a balanced PD. To construct the point cloud from a recorded oscillogram, we first employ a short-time Fourier transform with a window size equal to half of the chirping period. Obtained time-frequency plots for the target are presented in Fig. 4c. Time-frequency spectrograms contain 82k timeslices with typical 15 dB SNR for the target. We remove points with SNR below 10 dB from the point cloud. Noticeable reflection at 27 MHz in Fig. 4c is due to the reflection from the collimator. Peaks at 40–46 MHz offsets correspond to the target scene. We find a peak with maximal spectral amplitude in the time-frequency plot for each timeslice. The frequency of the peak provides the distance

information (radial coordinate) for each timeslice. Figure 4d provides a histogram of distance distribution for the target point cloud. We used zero-padding to increase FFT window size four times and to obtain a continuous distribution of radial distance. The point cluster at 9 m distance corresponds to the Styrofoam donut, 9.35 m to the PC monitor, and 9.7 m to the back wall. A small cluster of points at 9.6 m distance corresponds to the soldering station on the table. All peak widths are limited by the fundamental resolution of 12.5 cm. Polar and azimuthal coordinates were retrieved from the galvo scanner mirrors' driving signals, which were recorded on the same DSO. Interactive code for LiDAR data processing can be found in the code availability section. Figure 4e,f,g shows the point cloud of the scene with distance-based colouring, the donut is depicted in blue, PC monitor in green, and part of the wall in yellow. The scene has been properly reconstructed, although the dual layering of objects is visible in Fig. 4g due to discretization from 12.5 cm LiDAR resolution (no zero-padding was used in FFT).

To increase optical frequency excursion limited in our method and particular $Si_3N_4$ sample by injection locking range and improve LiDAR resolution, one might consider a different tuning scheme where the laser diode current and the piezoactuator voltage are synchronously tuned in a way that the laser is kept in injection-locked state. In this case, a diode current tuning might not be precisely linear as self-injection locking implies and preserves the linearity of cavity resonance tuning by the piezoelectrical actuator. In such a feed-forward scheme, the laser tuning range would be limited by piezoactuator's tuning range. Another option might be to use a two microring-based Vernier configuration for the laser[48,49], with the fast tuning provided by integrated piezoactuators.

**Hybrid integrated laser with fast tuning based on integrated low voltage PZT piezoactuator**. We investigate the reduction of tuning voltage requirement by introducing an integrated piezoelectric actuator based on lead zirconium titanate (PZT)[50,51]. The actuator has PZT as the main piezoelectric material and platinum (Pt) as the top and the bottom (ground) electrodes, as shown in Fig. 5a. The key difference to AlN is that the PZT process has a patterned ground plane to eliminate bond-pad capacitance, since the relative permittivity of PZT is >800 compared to ~9 for AlN. Using disk-shaped PZT actuator on top of $Si_3N_4$ microresonator with 100 GHz FSR (see Fig. 5b), we perform the heterodyne beat experiment with a free running ECDL reference laser. The DFB is self-injection locked at 240 mA driving current to the $Si_3N_4$ cavity resonance. By applying a triangular voltage ramp with an 81 kHz frequency and an amplitude from 0.5 V to 3.5 V and positive bias of 3.5 V, we measured optical frequency excursion from 230 MHz to 1832 MHz correspondingly, and fit a tuning efficiency of 520 MHz/V (see Fig. 5c). Figure 5d, e presents the laser frequency spectrogram and the corresponding tuning nonlinearity at 500 kHz ramp frequency with 1.1 V applied, confirming the PZT actuator nonlinearity 0.95% over 525 MHz frequency excursion.

**Discussion**
We have demonstrated a hybrid integrated laser with low noise while exhibiting up to 10 MHz flat laser frequency actuation. This is achieved using piezo-electrical actuators in conjunction with suppression of mechanical modes of the chip via apodization. The approach is based on foundry-ready processes that include photonic integrated circuits based on $Si_3N_4$ as well as AlN and PZT MEMS processing, and is therefore amenable to large-volume manufacturing. The combination of narrow linewidth (kHz level) along with the fast and flat actuation response, makes the source ideally suited for medium to long-range coherent LiDAR, as

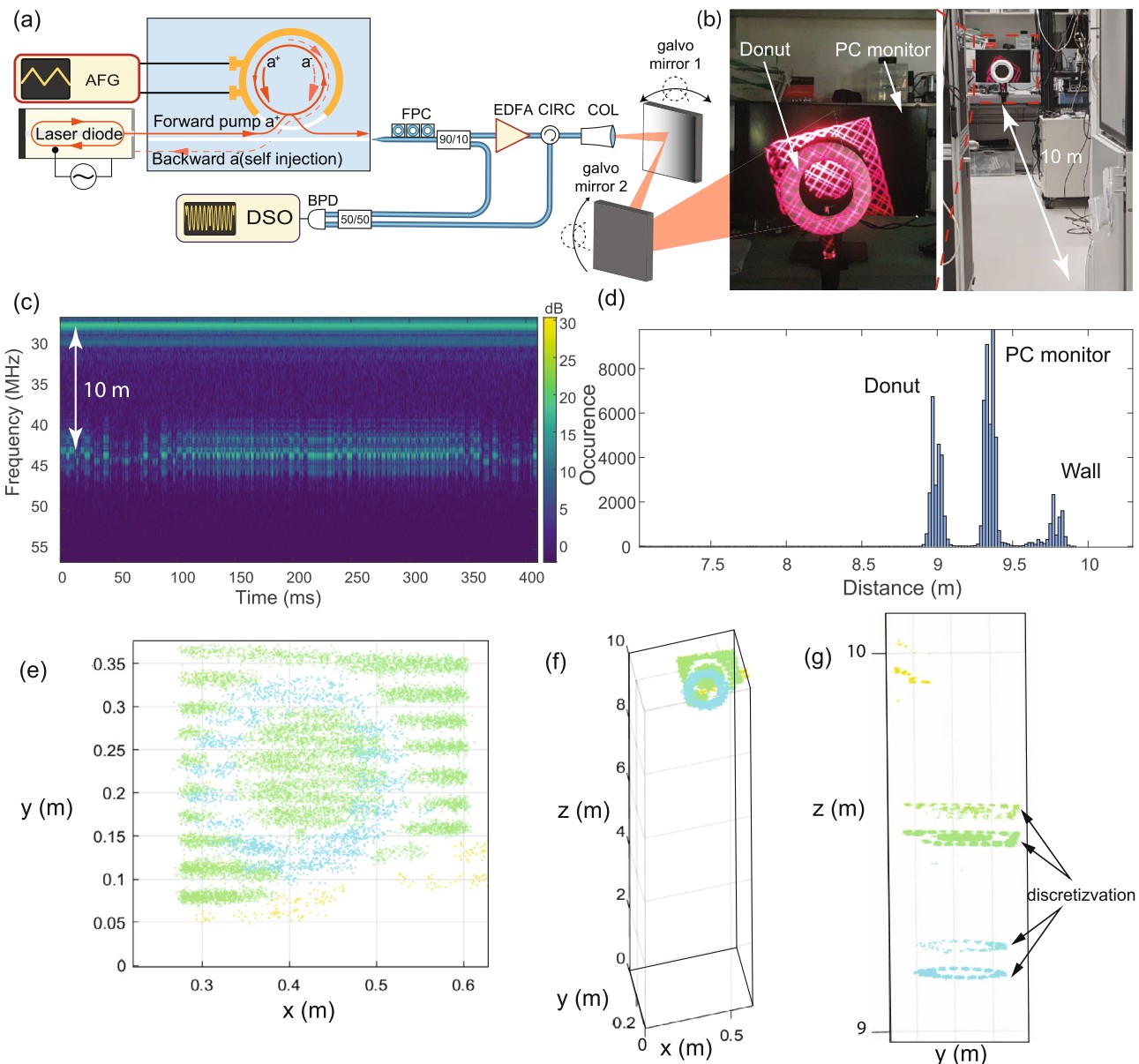

**Fig. 4 Optical ranging using the frequency-agile, hybrid integrated laser. a** Schematic of the setup for FMCW LiDAR measurement. A triangular ramp with 150 V peak-to-peak amplitude at 100 kHz rate is applied to the AlN piezoactuator providing a 1.2 GHz optical frequency excursion of the self-injection locked DFB laser. Beam steering is realised using a mechanical galvo scanner with two mirrors. **b** Photos of the target - a donut in front of a PC monitor. **c** Time-frequency plot for a signal from the target. **d** Histogram of distance distribution in the point cloud. **e–g** Point cloud of the target from different perspectives using a beam scanning pattern with 3 Hz vertical and 60 Hz horizontal triangular scanning frequencies. Point colours are based on distance. Donut corresponds to blue, PC monitor - green, wall - yellow.

required for autonomous driving, drone navigation, or industrial and terrain mapping. The combination of low noise and fast on chip tuning, alleviates the need for external components such as AOMs or single sideband modulators for fast frequency actuation, and may also find use in other areas such as locking of lasers to reference cavities, or atomic transitions - where a tight lock is required, and can be achieved with a high actuation bandwidth as demonstrated here. In addition, while our lasers were demonstrated at 1556 nm, the centre wavelength can be readily extended to other ranges, including the near IR and mid-infrared, due to the transparency of $Si_3N_4$. To further improve the performance of the laser system here, one can use laser chips with higher output power. Also, microresonators with larger mode volumes can be used to reduce the fundamental thermo-refractive noise. In addition, with careful design of the geometry of the piezoactuator

and the dimension of the $Si_3N_4$ photonic chip to suppress HBAR modes, the linear actuation bandwidth could extend into the GHz regime - limited only by the internal modes of the piezoactuator[31]. Last, viewed more broadly, our results show that low noise hybrid integrated photonic lasers based on the $Si_3N_4$ - as pioneered by Boller et al. - can now achieve unique performance metrics in terms of combining linearity, frequency agility, and phase noise. In many cases our laser supersede commercial bulk external-cavity diode lasers and fibre lasers, today's workhorses of distance metrology, spectroscopy and quantum optics. Detailed comparison with commercial ECDL and fibre lasers as well as many compact and photonic integrated tunable lasers based on monolithic InP chips is given in Table II of the supplementary information. Our results indicate the significant potential of hybrid integrated photonics with low loss dielectric

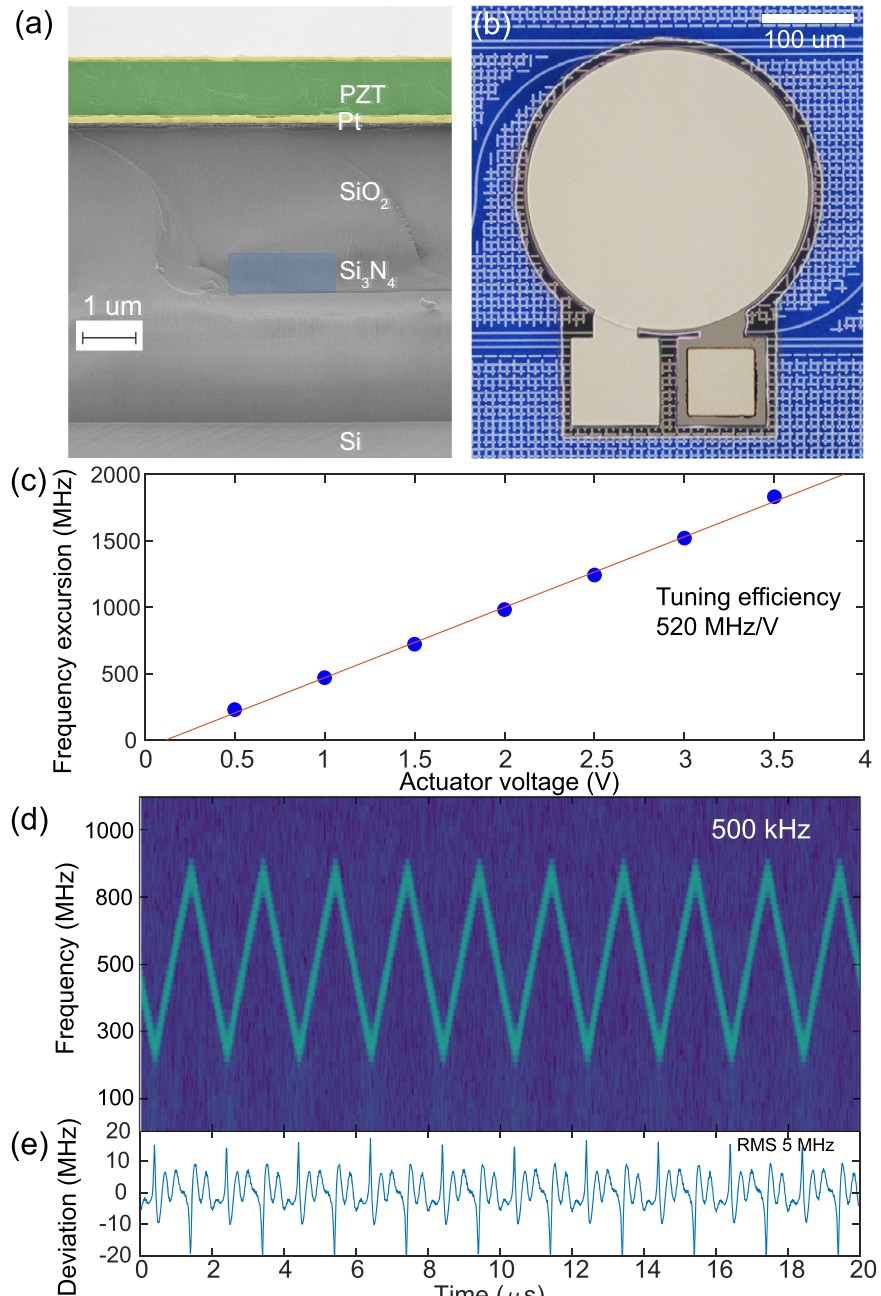

**Fig. 5 Low voltage fast tuning with integrated PZT piezoactuator. a** False-coloured SEM image of the sample cross-section, showing the PZT actuator integrated on the Si$_3$N$_4$ photonic circuit. The piezoelectric actuator is composed of Pt (yellow), PZT (green) layers on top of Si$_3$N$_4$ (blue) buried in SiO$_2$ cladding. **b** Optical micrograph of disk-shaped PZT actuator on top of Si$_3$N$_4$ microring with 100 GHz FSR. **c** Frequency excursion as a function of the voltage applied to the PZT actuator, measured at 81 kHz triangular chirping rate. Linear fit (red) provides 520 MHz/V tuning efficiency. **d** Time-frequency spectrogram of the heterodyne beat-note for 500 kHz triangular chirp repetition frequency with 1.1 V applied to the PZT actuator. **e** Residual of least-squares fitting of the time-frequency trace with symmetric triangular chirp pattern.

feedback and fast non-thermal actuation to replace decades-old prevailing technologies based on bulk fibre laser and bulk grating-based external cavity lasers.

## Methods

**Thermorefractive noise calculation**. To calculate TRN limit, we use the following expression for the effective temperature fluctuations[43,52]:

$$S_{\delta T}(\omega) = \frac{k_B T^2}{\sqrt{\pi^3 \kappa \rho C \omega}} \sqrt{\frac{1}{2p+1}} \frac{1}{R\sqrt{d_r^2 - d_z^2}} \frac{1}{[1+(\omega\tau_d)^{3/4}]^2}, \quad (1)$$

where $R$ is the microring resonator radius; Si$_3$N$_4$ material parameters $\rho = 3.29 \times 10^3$ kg m$^{-3}$ is density; $\kappa = 30$ W m$^{-1}$ K$^{-1}$ is thermal conductivity; $C = 800$ J kg$^{-1}$ K$^{-1}$ is specific heat capacity; $T = 300$ K, $d_z = 1.5$ μm and $d_r = 0.75$ μm stand for halfwidths of the fundamental mode, with orbital number $l$, azimuthal number $m$ and meridional mode number $p = l - m$, $\tau_d = \frac{\pi^{1/3}}{4^{1/3}} \frac{\rho C}{\kappa} d_r^2$.

**FMCW laser ranging experiment**. The DFB is directly butt-coupled to a 190.7-GHz FSR microresonator on a regular non-apodized chip placed on a carbon tape. The chip features a single AlN disk actuator. We employ laser frequency tuning by keeping the laser diode's current fixed, the laser self-injection locked to a Si$_3$N$_4$ resonance and by tuning only the cavity resonance with AlN piezoactuator. Tri-angular ramp with 100 kHz frequency from an arbitrary waveform generator is

amplified to 150 V (peak-to-peak amplitude) with a high voltage amplifier with 5 MHz bandwidth. The diode current is set near the centre of the injection locking range (281 mA for the particular resonance used) and AlN voltage is adjusted to keep the laser inside the self-injection locking range, resulting in a 1.2-GHz optical frequency excursion, corresponding to a FMCW LiDAR resolution of approximately 12.5 cm. While not required in actual operation, a small portion of the laser power was split of (95/5) to a fibre-coupled Mach-Zehnder interferometer for calibration of the frequency excursion. The length of the calibration MZI was measured with a frequency comb calibrated tunable diode laser scan. No active linearization or k-point sampling was necessary due to the excellent linearity and negligible hysteresis of the integrated AlN tuner. While 10% of the total laser power of 1.5 mW is used as a local oscillator (LO), 90% of the light is amplified by an EDFA to 10 mW and sent to a target through a collimator with an aperture of 8 mm, which was adjusted to fit the target range of 10 m. A double-axis galvanometric mirror scanner (Thorlabs GVS112) was used for the beam steering. Vertical and horizontal mirrors were rotated with constant speed at rates 3 Hz and 60 Hz correspondingly to cover the full scene during the measurement time of 400 ms.

## Data availability

Data used to produce the plots within this paper is available at https://doi.org/10.5281/zenodo.6328345 All other data used in this study are available from the corresponding author upon reasonable request.

## Code availability

Code used to produce the plots within this paper is available at https://doi.org/10.5281/zenodo.6328345.

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

## Acknowledgements

This work was supported by funding from the European Union H2020 research and innovation programme under FET-Open grant agreement no. 863322 (TeraSlice) and the Marie Sklodowska-Curie IF grant agreement No. 101033663 (RaMSoM). This material is based upon work supported by the Air Force Office of Scientific Research under award

number FA9550-19-1-0250. This work was supported by funding from the Swiss National Science Foundation under grant agreement No. 192293, No. 176563 (BRIDGE) and No. 201923 (Ambizione). This work was supported by funding from the European Space Technology Centre with ESA Contract No. 4000135357/21/NL/GLC/my. This work was also supported by the United States' National Science Foundation under grant No. PHY 18-39164, DMR 17-47426, and QIS DCL 20-063. The chip samples were fabricated in the EPFL centre of MicroNanoTechnology (CMi), and in the Birck Nanotechnology Center at Purdue University. AlN deposition was performed at OEM Group Inc. PZT actuators were fabricated at Radiant Technologies Inc. We acknowledge Lin Chang and John Bowers for providing the DFB laser.

## Author contributions

G.L. and J.R. performed the experiments with the help from Vy.S.; W.W. developed a theoretical model and performed numerical simulations. A.S. performed mechanical response measurements. J.L., R.N.W. and J.H. designed, fabricated and characterised the $Si_3N_4$ chips. H.T. designed and fabricated the AlN actuators. R.N.W. performed chip apodization. Vl.S. and A.V. performed actuator's bandwidth requirement simulations. G.L, J.R., A.S. and V.S analysed the data. W.W., G.L. and J.R. wrote the manuscript with input from all authors. T.J.K. and S.A.B. supervised the project.

## Competing interests

The authors declare no competing interests.
