## [Peer review file · Nature Communications]

Low-noise frequency-agile photonic integrated lasers for coherent rangingREVIEWER COMMENTS

Reviewer #2 (Remarks to the Author):

Review of NCOMM-21-43922-T

This is the third review of this paper. See below for comments on the most recent reply.

1. The paper title and abstract have been adequately changed.
2. If the paper is given the original submission date, and if the PZT results were added during the review process after the original submission date, then I stand firm on my review that these results should not be included. The PZT results are from a vendor who did all of the processing, and if the results were added after the first submission date, they are not results that would stand on their own. While the authors may be adamant about including these results, I stand firm on my point as a reviewer with respect to this issue. Of course the Editor will have the final say.
3. The common definition of ultra-low phase noise is generally accepted in the literature, although not defined in the community, and the reviewer appreciates the author's acknowledgment of the values given in the review.
4. It is true, that the large body of work on fast tuning and switching generally involved mode- hopping.
5. The comparison to InP lasers is now much improved.
6. The focus on LIDAR is much improved.
7. The use of large signal modulation is different than generally accepted. Large signal modulation involves a step response. In this paper a triangle wave is used, that is generally called a linear sweep. The bandwidth in the triangle and step (or square) is very different as am sure the Authors know. This point needs to be updated so that small signal and large signal are better aligned with generally used practices.
8. Wafer-scale manufacturable is not open to interpretation. It means that one can process the laser or some part of the laser on a 200mm or larger substrate. Back end processing and hybrid assembly can be seen as wafer-scale compatible if the base devices are demonstrated at 200mm or larger. The back-end assembly will dominate the yield and cost in any case, as well as testing an calibration. CMOS compatible means that the wafers can be run in a foundry line that also handle pure CMOS and the line operators will allow such a process to run on that line. These are well known definitions and the authors need to adhere to this.
9. Again, the use of broadband and narrow band needs to be specific. Modulation with a 1 GHz sinusoid is different than 1 GHz modulation bandwidth, which in the digital world means support from some low frequency out to a 3dB 1 GHz without any resonance peaks. The authors must make sure all language is consistent and does not leave the reader to mistake the claims made about modulation bandwidth. To be unambiguous it is suggested that the term narrow-band modulation of up to 1GHz is demonstrated, this is how the RF communities will specify such modulation. The authors need to carefully go through the text and make sure it is unambiguous.
10. The authors need to be clear about when they are talking about modified ST linewidth vs. integral (which includes technical). Please be very specific every time it is mentioned and then this issue will be taken care of.
11. The following two comments are with regard to the author's interaction in the review process, which for this manuscript at times has boarder lined on unpleasant. I understand the reviewer might be frustrated, but it is not ok to take it out on the reviewer the way they have chosen to:
 - a. Using comments like "Again, the Referee seems to have not studied the manuscript with great care." Is really not to the benefit of the author and really just get in the way of a productive review process. The authors have many other ways to vent and voice their

frustration, in manners that can be more professional. Seeing statements like this really make it difficult to review such a manuscript and do so for so many hours. More professional replies to the reviewer would be greatly appreciate, I am just doing my job.

b. It is too bad the authors do not appreciate the amount of time that has gone into reviewing their manuscript. While I can understand that the authors might be upset that it was stated that they are not known experts in the field of communication, which is true. It is unfortunate they took the review and the fact that they are not experts in a certain area as personal.

c. At the same time, to say that the reviews were "condescending" and the other comments along about the reviewer, are not appropriate under any circumstances. It creates a hostile environment that no reviewer wants to be part of and does not show a willingness of the authors to work with the reviewer to make a better paper, which is an important part of the purpose of this process.

d. I really recommend that the authors tone-down their response to people doing voluntary work and trying to maintain quality of the Nature review process, and view it as part of the process.

Reviewer #3 (Remarks to the Author):

The authors present a self-injection locked diode laser in an integrated package via backscattering of light in a high-Q microring resonator. The frequency of the injection-locked diode laser can be tuned by tuning the resonant frequency of the microring. For this the piezo-electric effect is used (either in AIN or PZT). Due to the fast response time of the piezo-electric effect, fast and linear tuning of the laser frequency is obtained. Finally, the high Q of the microring resonator significantly reduced the frequency noise of the laser to a low value at high offset frequencies. At intermediate offset-frequencies thermorefractive noise is dominant while long term stability is not reported. The authors use this laser in a LIDAR demonstration.

The integrated laser itself combines several features that have been demonstrated separately and for the first time demonstrates agile and, in particular, fast linear tuning of the laser wavelength. The authors start with presenting an extensive study on the optical and acoustic properties of the integrated laser. The latter are important for linear tuning of the wavelength.

The authors have put a large effort in determining the intrinsic linewidth, however the authors do not make clear which offset frequencies in the noise are, or which linewidth is, of relevance for the application of coherent ranging. Is this the FWHM, and then determined over what time interval? The authors only make a vague statement that a laser with a low phase noise is of high interest for coherent ranging. The ranging experiment is done using a high-Q microring resonator with an FSR of almost 200 GHz, which means that the thermorefractive noise is higher than for the rings with a lower FSR (longer rings, so larger mode volume). By following the authors arguments, the ring with the smallest FSR should have the lowest noise and therefore be best suited for coherent ranging. Why is the ring with FSR of 200 GHz used? Further, the authors present coherent ranging at a distance of 10 m with a resolution of 12.5 cm. The beat frequency at this range is about 45 MHz, 1% of which is 450 kHz. So why is a very low noise frequency required?

Finally, no comparison is made with the state-of-the-art in coherent ranging. The reader has therefore no clue how to position this work in relation to other work on light sources for coherent ranging.

Overall, I find the internal consistency of the paper lacking in that choices are made but not explained, and that it is unclear what requirements coherent range put on the light source (e.g., for different ranges).

More detailed comments:

1. Figures should be mentioned in the order that they are discussed in the text (or vice versa). E.g. Fig 1d is discussed before 1b,c. Figure 2c is discussed before 2a,b.
 2. References 1 and 2 are used for very similar statement. Ref 1 is from 2012 while ref 16 is from 2020. The abstract of ref 1 mentions that most promising techniques will be discussed for future data centers. I expect that this manuscript does not yet discuss or mention application in data centers on a commercial scale. So, in both cases only Ref. 2 should be used.
 3. "thermo-dynamical noise, such as thermo-refractive noise due to refractive index fluctuations, constitutes another fundamental limit". I find this a bit misleading, as there can only be one fundamental limit, and that is the one given by spontaneous emission. This fundamental limit will become apparent when all other noise sources are reduced to below this level. At sufficiently high offset frequencies also thermorefractive noise will become lower than the quantum noise. What is true is that at intermediate offset frequencies, thermorefractive noise is dominant over quantum noise and becomes the limiting noise source. However, it is not the fundamental limit as it can be lowered by lowering the temperature and increasing the mode volume.
 4. When discussing the lowest phase noise obtained with injection locking, I am missing a reference to the recent work of Bowers, doi: 10.1364/OL.439720.
 5. "Applying the technique of self-injection locking to ultra low loss ...". The hybrid integrated diode lasers pioneered by the group of Boller is not based on self-injection locking. Self-injection locking requires a laser that is capable of laser oscillation on its own. That is not the case with the gain sections used in the hybrid integrated diode lasers. The silicon nitride feedback circuit is essential for laser oscillation. In contrast, the work discussed here, and, e.g., presented recently by the group of Bowers and the group of Lipton, uses either DFB or FP diode lasers that are fully functioning diode lasers by themselves.
- Furthermore, in the construct "Boller et al." the first author of the article is used, not the last one. I think the authors should use "pioneering work of the group of Boller ...". This appears again in the conclusions.
6. "we only consider linear regime ..." => "we only consider the linear regime ..."
 7. "(cf. Fig. 1(f)) while maintaining injection locking." The last part about maintaining injection locking is already mentioned in the first part of the sentence.
 8. A tuning range of up to 2 GHz is reported. Compared to other techniques this is a relatively small value, e.g., single sideband generation would easily allow tuning ranges between 10 to 20 GHz. Can this tuning range be increased and what would be needed for that?
 9. "which has only been shown in Si₃N₄ low-confinement waveguides [3]". A better result exists since ref. 3 has been published, see comment 4.
 10. The main text reports a value for the FWHM linewidth for the 10-GHz FSR microring resonator while the 200-GHz FSR microring is used in the coherent ranging experiment. The FWHM should be reported for injection locking using the 200 GHz microring as that value is of interest for the coherent ranging experiment.
 11. Figure 6 in SI shows that the frequency noise below 1 kHz flattens, even for the free-running DFB laser. Why is this? If 1/f noise is dominant in this region, I would expect an increase of the frequency noise for smaller offset frequencies.
 12. Why is there no information given on the accuracy of the measured single sided PSD?
 13. "showcases the remarkable frequency agility of our system." Indeed a good performance is demonstrated, however, similar if not better performance can be obtained

using single sideband frequency generation followed by injection-locked amplification in a Bragg diode laser, which, in principle, could also be implemented on a single chip assembly. The authors need to put their work in perspective and discuss state-of-the-art.

14. "Together with more than 1 GHz tuning range, such a high actuation bandwidth exceeds the performance of common benchtop laser systems that rely on bulk piezo or electrooptic components," This statement is not true, see, e.g., doi: 10.1109/JLT.2021.3050772

15. Reference 44 is not a proper reference. This reference does not discuss the properties of the laser and is just a conference abstract which promises a lot without showing details. Better would be to use "P. Feneyrou, et al., Appl. Opt. 56(35) 9663-9675 (2017)" and "P. Feneyrou, et al., Appl. Opt. 56(35) 9676-9685 (2017)" of the same authors, which are regular articles providing more and detailed information.

16. The abbreviation FEM is not defined in the text.

17. "We apply triangular chirp with 100 kHz frequency to ...". First it should be "We apply a triangular chirp ...". Further, this is two orders of magnitude smaller than what can be realized by the laser (as demonstrated earlier in the manuscript). Why is the full potential not used, i.e., why go through all the effort of showing a flat actuation response to high frequency if it is not used in the application?

18. "Obtained time-frequency plots are presented in Fig. 4(c) for the target and in Fig. 4." Some editing issues from the previous version of the text?

19. "See attached to the SI documented interactive code for LiDAR data processing". Improve grammar.

20. "No zero-padding was used in processing point cloud data". Are figures 4d and 4(e-t) not different representations of the same data. Why is different data processing used for these figures (with and without zero-padding)

21. "alleviates the need for external components ... single sideband modulators ...". on-chip single sideband modulators have been available for a long time, see, e.g., doi: 10.1109/ICTON.2015.7193321

22. In the conclusions the authors mention that the source is ideal for coherent ranging applications (among others). However, to operate in the linear regime, the output power is limited to 1.5 mW and external amplification was needed for the coherent ranging test. What would happen in the integrated laser was operated in the nonlinear regime? What changes and how would this affect the application? Do the authors foresee that this technology would allow for coherent ranging without additional amplification, i.e., can the self-injection locked laser provide sufficient power for the application by itself?

In conclusion, I cannot recommend publication of the manuscript in its current form.

REVIEWER COMMENTS

We are grateful that three Referees have again seen our manuscript. We appreciate reviews by the reviewers and have modified the manuscript to highlight changes we made to address concerns of Reviewer #2 and Reviewer #3. Here, we present a point-by-point reply (in **blue**) to the reviewers' comments (in **black**), as well as the action taken (in **red**) and new/rephrased sentences in the main manuscript (in **green**).

Sincerely Yours,

Tobias J. Kippenberg

Reviewer #2 (Remarks to the Author):

Review of NCOMM-21-43922-T

This is the third review of this paper. See below for comments on the most recent reply.

We thank again Referee #2 for his/her effort reviewing our manuscript and appreciate the amount of his/her time that has gone into three rounds of reviewing the manuscript. We also thank the Reviewer for maintaining a high quality of the review process for the Nature Portfolio. We express our willingness to work productively with the Reviewer to make a better paper. We regret that some of our previous comments sounded not appropriate for the Reviewer. This was not our intention under any circumstances. However, we would like to note that some aspects and criticism of the review we perceived as unfair and the data removal request as unusual.

1. The paper title and abstract have been adequately changed.

We thank the Referee for this acknowledgment.

2. If the paper is given the original submission date, and if the PZT results were added during the review process after the original submission date, **then I stand firm on my review that these**

results should not be included. The PZT results are from a vendor who did all of the processing, and if the results were added after the first submission date, they are not results that would stand on their own. While the authors may be adamant about including these results, I stand firm on my point as a reviewer with respect to this issue. Of course the Editor will have the final say.

First, we would like to clarify that to the PI, it is the first time to have received a request to **remove data** from the review that was explicitly asked.

Let us clarify: The PZT results were added to the main manuscript during the first round of reviews in Nature Photonics. PZT fabrication was performed by an external vendor (Radiant Technologies Inc.). However, the actuator design, placement optimization, and all characterization and testing were performed by the authors. We don't think that external foundry – common in integrated photonics – should exclude or diminish the result. As an aside: the work cited by the Referee (Jin et al., Nat. Photon. 15 346 (2021)] is using an external foundry (Towers Semiconductor).

The question about increasing voltage-to-frequency conversion efficiency was raised explicitly in the first round of review by Referee #2 (comment 13) and by Referee #1. Here we quote the original comments:

“Also, agile tuning of the cavity using 120V voltage is challenging. Could the authors comment on improvement of the voltage sensitivity of the system.” (Referee #1)

“13. The reported AlN tuning voltage is 125 Vpp, which is pretty large. However, throughout the manuscript, the authors have not commented on the limitation of this large tuning voltage. Can the authors explain why it requires such high voltage, and what could mitigate it?” (Referee #2)

As for ourselves: We would like to keep our demonstration of an integrated actuator based on PZT material, which reduces the voltage requirement for the laser system to a level of <10V, in the manuscript.

We agree that the Editor should decide.

3. The common definition of ultra-low phase noise is generally accepted in the literature, although not defined in the community, and the reviewer appreciates the author's acknowledgment of the values given in the review.

We thank the Referee for this acknowledgment.

4. It is true, that the large body of work on fast tuning and switching generally involved mode-hopping.

We thank the Referee for the acknowledgement of the fact that our platform surpasses most of the lasers in mode-hop-free tuning speed.

5. The comparison to InP lasers is now much improved.

We appreciate that the Reviewer gave a positive evaluation of the extended comparison table and revised text.

6. The focus on LIDAR is much improved.

We appreciate that the Reviewer gave a positive evaluation of the revised manuscript.

7. The use of large signal modulation is different than generally accepted. Large signal modulation involves a step response. In this paper a triangle wave is used, that is generally called a linear sweep. The bandwidth in the triangle and step (or square) is very different as am sure the Authors know. This point needs to be updated so that small signal and large signal are better aligned with generally used practices.

Please refer to our answer to comment #9 below.

8. Wafer-scale manufacturable is not open to interpretation. It means that one can process the laser or some part of the laser on a 200mm or larger substrate. Back end processing and hybrid assembly can be seen as wafer-scale compatible if the base devices are demonstrated at 200mm or larger. The back-end assembly will dominate the yield and cost in any case, as well as testing an calibration. CMOS compatible means that the wafers can be run in a foundry line that also handle pure CMOS and the line operators will allow such a process to run on that line. These are well known definitions and the authors need to adhere to this.

We appreciate the Reviewer's comment though we do not share that "Wafer-scale manufacturable" refers to laser manufacturing at 200 mm or larger – the wafer size should be immaterial. This appears to exclude InP, which is typically processed commercially based on 2, 3 or 4 inch wafer level (e.g. from **Seminex Corp.**, <https://www.seminex.com/wp-content/uploads/2021/06/SemiNex-workmanship-std-app-notes-rev-4-English-Final.pdf>).

We note that Si₃N₄ can since recently be processed at 8-inch wafers (e.g. from **XFAB/Ligentec**).

Nevertheless, we have complied with the Reviewer's request and changed our terminology. We entirely agree that back-end assembly will dominate yield and cost – and thus have avoided giving the impression that the entire process is "wafer-scale manufacturable". We have removed this.

Action taken: We removed the term "wafer-scale-manufacturing-compatible" from the abstract and replaced it with the more general term: foundry-based technology. We removed the term "CMOS-compatible" Si₃N₄ from the main manuscript.

Rephrased sentence: "Our approach uses foundry-based technologies - ultralow-loss (1 dB/m) Si₃N₄ photonic microresonators ..."

9. Again, the use of broadband and narrow band needs to be specific. Modulation with a 1 GHz sinusoid is different than 1 GHz modulation bandwidth, which in the digital world means support from some low frequency out to a 3dB 1 GHz without any resonance peaks. The authors must make sure all language is consistent and does not leave the reader to mistake the claims made about modulation bandwidth. To be unambiguous it is suggested that the term narrow-band modulation of up to 1GHz is demonstrated, this is how the RF communities will specify such modulation. The authors need to carefully go through the text and make sure it is unambiguous.

We thank the Reviewer for this comment. We double-checked all uses of the word “bandwidth” in the manuscript and made sure that it only indicates the electrical modulation bandwidth (i.e. flat response up to a certain frequency). The term ”narrowband modulation” however, describes a specific format of FM modulation, where the frequency deviation is smaller than the symbol rate, which means that the total phase deviation is smaller than 2π . This implies that the modulated spectrum measured over many symbols does not very much differ from the unmodulated spectrum. For our FMCW LiDAR, the modulated spectrum is about 100000 times wider than the CW spectrum of the laser. Hence the term narrowband modulation is not an appropriate description of the chirp frequency modulation formats commonly used in FMCW LiDAR. There is also no appropriate definition of a symbol rate for FMCW LiDAR. The relevant phase information in FMCW LiDAR is spread across the whole laser chirp of 5 us and recovered by Fourier transform. The total phase deviation for one chirp ramp is on the order of 250000 rad. Hence we believe that calling it narrowband modulation would raise more confusion with the average reader than it would solve. It is our opinion that using established terminology of the coherent laser ranging community is the best choice for this manuscript [e.g. Behroozpour et al. (2017)], also given the new title of the manuscript that highlights the potential use of our laser system for coherent laser ranging.

Action taken: We clarify the use of the term bandwidth throughout the paper not to conflate optical and electrical bandwidths. We replace the term “large actuation bandwidth B (which determines LiDAR resolution $c/2B$)” with “large optical frequency excursion B (which determines LiDAR resolution $c/2B$)”.

10. The authors need to be clear about when they are talking about modified ST linewidth vs. integral (which includes technical). Please be very specific every time it is mentioned and then this issue will be taken care of.

We amended the manuscript to be very clear and comprehensive about laser linewidth definitions for all three tested systems.

Action taken: We state the intrinsic laser linewidth and integrated laser linewidth, including the integration time for all three tested systems in the main manuscript.

New sentence: “The full width at half maximum linewidth, which is calculated by integration of the frequency noise from the beta-line to the inverse integration time, is for the 9.87 GHz FSR Si_3N_4 device 7.5 kHz at 1 ms integration time, 18.7 kHz at 10 ms, and 21.5 kHz at 100 ms. For the 190.7 GHz FSR Si_3N_4 device the integrated linewidth is 43.27 kHz at 1 ms integration time, 73.7 kHz at 10 ms, and 81.7 kHz at 100 ms. For the 2.45 GHz FSR Si_3N_4 device the integrated linewidth is 14 kHz at 1 ms integration time, 127 kHz at 10 ms, and 130 kHz at 100 ms (cf. SI Fig. 6)”

11. The following two comments are with regard to the author’s interaction in the review process, which for this manuscript at times has boarder lined on unpleasant. I understand the reviewer might be frustrated, but it is not ok to take it out on the reviewer the way they have chosen to: a. Using comments like “Again, the Referee seems to have not studied the manuscript with great care.” Is really not to the benefit of the author and really just get in the way of a productive review process. The authors have many other ways to vent and voice their frustration, in manners that can be more professional. Seeing statements like this really make it difficult to review such a

manuscript and do so for so many hours. More professional replies to the reviewer would be greatly appreciate, I am just doing my job.

b. It is too bad the authors do not appreciate the amount of time that has gone into reviewing their manuscript. While I can understand that the authors might be upset that it was stated that they are not known experts in the field of communication, which is true. It is unfortunate they took the review and the fact that they are not experts in a certain area as personal.

c. At the same time, to say that the reviews were “condescending” and the other comments along about the reviewer, are not appropriate under any circumstances. It creates a hostile environment that no reviewer wants to be part of and does not show a willingness of the authors to work with the reviewer to make a better paper, which is an important part of the purpose of this process.

d. I really recommend that the authors tone-down their response to people doing voluntary work and trying to maintain quality of the Nature review process, and view it as part of the process.

We all agree with the Reviewer that none of us wants to take part in a hostile review environment, neither as authors nor as reviewers – and the open review process that publishes reviews in Nature Communications is an aspect we firmly believe in. We assure the Reviewer that we take the review process very seriously regardless of whether we take part as authors or reviewers. The authors collectively review around 20 journal manuscripts each year, frequently also for *Nature Communications*. We believe that the peer review process is one of the pillars of scientific accountability and the scientific method of conjecture and verification. We immensely appreciate the time and effort spent by the reviewers of our manuscripts, especially in the case of multi-round review processes.

We also immensely appreciate any comment that is intended to improve the manuscript and comments done in “good faith”. We frequently observe that our manuscript improves in quality during the review process and we believe the same to be the case here. We apologize for the comments in our previous reply that were received as unprofessional and not appropriate.

That said, it should also be noted that our replies have taken us in response major time, and from the time of submission until now, almost a full year has passed. Moreover, our replies have been exceptionally carefully prepared to reach now a cumulative 60+ pages. To the authors, it is not pleasant and unfair if new points are brought up in each round.

The PI who supervised this work and is writing this reply would like to take the opportunity to explain ourselves on the matter:

- 1) It is indeed surprising to have seen comments during the review that my group has no expertise in coherent communications and to be criticized for text that was actually removed in our resubmission. I would like to mention that this comment is unjust, given that my team has collaborated since 2013 with Prof. Christian Koos and jointly published widely recognized work in the field of coherent communications: [Pfeifle, et al., *Nature Photon* 8, 375–380 (2014)]. and [Marin-Palomo P. et al., *Nature* 546, 274–279 (2017)].

- 2) Irrespective of this point, and while we have removed the comment, we offered the referee examples that prove that narrow linewidth lasers are useful for optical communications – it is a product that is commercially sold by NeoPhotonics (see footnote¹).
- 3) We would like to mention that the Reviewer raised several points in the second review (1st Appeal at Nature Photonics) that were referring to an outdated version of the manuscript or referring to terms not mentioned in the manuscript. In particular, application for carrier recovery in coherent communications has already been removed from the revised manuscript in the second round of review. Also, the frequency noise below 1 kHz presented in the SI was not taken into account in the first round of review. While we certainly believe that those were made in honest error, any author and Reviewer must understand the frustration of receiving such comments during the review. We have tried our best to comply with all points, including those where we had other opinions.

We thank the Reviewer for doing voluntary reviewing and making our paper better through the high-quality review process. We do acknowledge the significant time spend on the manuscript.

¹ https://compoundsemiconductor.net/article/113193/NeoPhotonics_Has_Shipped_2_Million_Ultra-Narrow_Linewidth_Lasers#:~:text=NeoPhotonics%20Ultra%2DNarrow%20Linewidth%20tunable,a%20total%20fiber%20capacity%20of

Reviewer #3 (Remarks to the Author):

We thank Referee #3 for his/her review of the revised manuscript and provided comments.

The authors present a self-injection locked diode laser in an integrated package via backscattering of light in a high-Q microring resonator. The frequency of the injection-locked diode laser can be tuned by tuning the resonant frequency of the microring. For this the piezo-electric effect is used (either in AIN or PZT). Due to the fast response time of the piezo-electric effect, fast and linear tuning of the laser frequency is obtained. Finally, the high Q of the microring resonator significantly reduced the frequency noise of the laser to a low value at high offset frequencies. At intermediate offset-frequencies thermorefractive noise is dominant while long term stability is not reported. The authors use this laser in a LIDAR demonstration.

The integrated laser itself combines several features that have been demonstrated separately and for the first time demonstrates agile and, in particular, fast linear tuning of the laser wavelength. The authors start with presenting an extensive study on the optical and acoustic properties of the integrated laser. The latter are important for linear tuning of the wavelength.

We thank the Reviewer for the report and appreciate that the novelty is recognized and emphasized of the ability to tune linearly, fast with narrow linewidth.

The authors have put a large effort in determining the intrinsic linewidth, however the authors do not make clear which offset frequencies in the noise are, or which linewidth is, of relevance for the application of coherent ranging. Is this the FWHM, and then determined over what time interval? The authors only make a vague statement that a laser with a low phase noise is of high interest for coherent ranging.

In the revised manuscript, we state the intrinsic laser linewidth and integrated laser linewidth using the beta line method, including the integration time for all three used lasers.

Laser phase noise limits the maximum operating distance and ranging precision in FMCW LiDAR. We added citations to [Behroozpour et al. (2017), Harris et al. (1998)] with analysis of laser coherence role in FMCW LiDAR and derivation of detector's photocurrent SNR depending on laser frequency noise level.

Please also refer to our simulations of the laser phase noise and chirp nonlinearity penalties of FMCW LiDAR detection presented below. We decided against their inclusion in the manuscript and the SI because we feel that they can be reproduced from textbooks easily.

Action taken: We added a citation to [Behroozpour et al. (2017) and Harris et al. (1998)] to the main manuscript and add a sentence discussing the implications of the linewidth and linearity for FMCW ranging.

New sentence: “Laser phase noise limits the maximum operating distance and ranging precision in FMCW LiDAR. However, a key requirement for FMCW at long range is in addition to low phase noise, frequency agility, i.e to achieve fast, linear and hysteresis-free tuning.”

The ranging experiment is done using a high-Q microring resonator with an FSR of almost 200 GHz, which means that the thermorefractive noise is higher than for the rings with a lower FSR (longer rings, so larger mode volume). By following the authors arguments, the ring with the smallest FSR should have the lowest noise and therefore be best suited for coherent ranging. Why is the ring with FSR of 200 GHz used?

Indeed, we used a microresonator with 190.7 GHz FSR for the LiDAR demonstration, and it has the largest frequency noise among all tested lasers (see Figure 2 of the main manuscript). We used a 190 GHz device because both 10 GHz and 2 GHz devices had no integrated piezoelectrical actuators on top.

Further, the authors present coherent ranging at a distance of 10 m with a resolution of 12.5 cm. The beat frequency at this range is about 45 MHz, 1% of which is 450 kHz. So why is a very low noise frequency required?

We agree that the noise performance of our lasers outperforms the requirements for 10m range FMCW LiDAR. The distance in the experiment was limited to 10m due to the size of our laboratory. To address the influence of the laser linewidth and chirp nonlinearity on FMCW LiDAR with longer range targets, we simulated a length measurement using linearly chirped FMCW laser with different Gaussian linewidth and chirp nonlinearity. For ranges beyond 100 m, linewidths substantially below 100 kHz and linearities below 0.1% are a must to avoid SNR and resolution penalties.

Figure 1: Simulations of the self-homodyne beatnote for 500 m distance for Gaussian laser linewidth 1 kHz and 10 kHz (Gaussian contribution) and different RMS tuning nonlinearity. Chirp rate is 10 kHz, chirp amplitude is 1 GHz.

In simulations, a chirped signal with linearly changing instantaneous frequency from 0 to 1 GHz during 50 us (corresponding to 10kHz chirp rate) was self-homodyned after the 500 m fiber delay line. We consider two types of deviation from the perfectly linear triangular in this simplified model. First, we added Gaussian noise to the instantaneous frequency with zero mean and

dispersion equal to the laser linewidth. Second, we added quadratic nonlinearity with different RMS parameters (normalized to the full 1 GHz chirp range) to the instantaneous frequency.

Finally, no comparison is made with the state-of-the-art in coherent ranging. The reader has therefore no clue how to position this work in relation to other work on light sources for coherent ranging.

The comparison table II in the SI was requested by the Referees and included both low-noise integrated lasers and swept lasers for LiDAR. The table was extended in the previous round of review.

Comparison table items C.V. Poulton et al.[29], N. Satyan et al.[30], X. Zhang et al.[31], M. Okano et al.[34], DiLazaro et al.[35] contain lasers for coherent ranging experiments.

We believe that a separate comprehensive comparison of all lasers for coherent ranging also with respect to other coherent ranging applications such as optical coherence tomography and optical frequency domain reflectometry, which have different requirements for chirp range, linearity, and laser linewidth is more suitable for a review paper and goes beyond the scope of this paper, which presents a new technology based on laser self-injection locking and piezoelectric frequency modulation of a high-Q optical microresonator.

Overall, I find the internal consistency of the paper lacking in that choices are made but not explained, and that it is unclear what requirements coherent range put on the light source (e.g., for different ranges).

To improve the consistency of the paper and motivate the choices made, we add citations to [Behroozpour et al. (2017) and Harris et al. (1998)]. Behroozpour et al. (2017) constitute an excellent tutorial on FMCW LiDAR with an analysis of most requirements, including laser noise and tuning linearity.

We added a simulation of piezoactuator bandwidth requirements to achieve the required linearity at a given chirp rate in the SI.

Action taken: We added citations to [Behroozpour et al. (2017) and Harris et al. (1998)] to the main manuscript to support our statements of the benefits of low noise lasers for FMCW LiDAR. We added sections to the SI on piezoactuator bandwidth requirement simulations.

New sentences in the main manuscript: “Recently, autonomous driving and areal mapping have increased the interest in such sources and a fully hybrid integrated low-noise, high and frequency-agile source could hence unlock further applications of coherent FMCW LiDAR. Laser phase noise limits the maximum operating distance and ranging precision in FMCW LiDAR. However, a key requirement for FMCW at long range is in addition to low phase noise, frequency agility, i.e to achieve fast, linear and hysteresis-free tuning.”

More detailed comments:

1. Figures should be mentioned in the order that they are discussed in the text (or vice versa). E.g. Fig 1d is discussed before 1b,c. Figure 2c is discussed before 2a,b.

We thank the Referee for pointing out the wrong order in the discussion.

Action taken: We reordered the discussion of mentioned Figures in the main text. We kept the same sentences but changed their order in section “Laser frequency noise measurements”.

2. References 1 and 2 are used for very similar statement. Ref 1 is from 2012 while ref 16 is from 2020. The abstract of ref 1 mentions that most promising techniques will be discussed for future data centers. I expect that this manuscript does not yet discuss or mention application in data centers on a commercial scale. So, in both cases only Ref. 2 should be used.

We thank the Referee for his comments on the relevance and the priority for mentioned references.

Action taken: We removed Ref. 1, 16 from the main manuscript

3. “thermo-dynamical noise, such as thermo-refractive noise due to refractive index fluctuations, constitutes another fundamental limit”. I find this a bit misleading, as there can only be one fundamental limit, and that is the one given by spontaneous emission. This fundamental limit will become apparent when all other noise sources are reduced to below this level. At sufficiently high offset frequencies also thermorefractive noise will become lower than the quantum noise. What is true that at intermediate offset frequencies, thermorefractive noise is dominant over quantum noise and becomes the limiting noise source. However, it is not the fundamental limit as it can be lowered by lowering the temperature and increasing the mode volume.

Given that operation temperature and volume are finite and cannot be decreased and increased indefinitely, any laser cavity will have a thermodynamical limit– that for a sufficiently high output power of the laser will always be larger than the modified Schawlow-Townes noise at lower frequency offsets. For photonic integrated waveguide resonators, which have an inherently small mode volume and very high optical quality factors, the relevant frequency where the TRN supersedes the ST noise level is in the kHz range and TRN becomes the fundamental limit relevant for the optical linewidth. Our use of the term fundamental is grounded on the fact that the TRN is directly derived from one of the most fundamental principles of statistical physics: the fluctuation-dissipation theorem. We also quote the title of M. Gorodetsky’s paper on the topic: “Fundamental thermal fluctuations in microspheres”.

Action taken: We rephrased the sentence to provide clarity around the term fundamental.

Rephrased sentence: “In addition to quantum noise, thermodynamical noise, such as thermo-refractive noise due to refractive index fluctuations, constitutes another limit”.

4. When discussing the lowest phase noise obtained with injection locking, I am missing a reference to the recent work of Bowers, doi: 10.1364/OL.439720.

Action taken: We included the citation of this work, which was published during the review process of our paper, in the abstract and the main text.

New sentence: “Using weak confinement Si₃N₄ waveguides and laser self-injection locking it has culminated recently in the demonstration of frequency noise of 0.006 Hz²/Hz at 4 MHz offset.”

5. “Applying the technique of self-injection locking to ultra low loss ...”. The hybrid integrated diode lasers pioneered by the group of Boller is not based on self-injection locking. Self-injection locking requires a laser that is capable of laser oscillation on its own. That is not the case with the gain sections used in the hybrid integrated diode lasers. The silicon nitride feedback circuit is essential for laser oscillation. In contrast, the work discussed here, and, e.g., presented recently by the group of Bowers and the group of Lipton, uses either DFB or FP diode lasers that are fully functioning diode laser by themselves. Furthermore, in the construct "Boller et al." the first author of the article is used, not the last one. I think the authors should use “pioneering work of the group of Boller ...”. This appears again in the conclusions.

We thank the Referee for pointing out this omission in the main text. We removed “self-injection locking” from a description of work of Boller’s group. We use the term “hybrid integrated lasers” based on Si₃N₄ feedback circuits.

Action taken: We rephrased the sentence using “pioneering work of the group of Boller ...”.

Rephrased sentence: “Using Si₃N₄ TriPleX waveguides as first demonstrated by the pioneering work of the group of K.J. Boller, has enabled hybrid integrated lasers with Hz-level Lorentzian linewidth, that have shown steady improvements.”

6. “we only consider linear regime ...”=> “we only consider the linear regime ...”

We thank the Referee for pointing out this grammar mistake.

Action taken: We added the article.

7. “(cf. Fig. 1(f)) while maintaining injection locking.” The last part about maintaining injection locking is already mentioned in the first part of the sentence.

We thank the Referee for pointing out this mistake.

Action taken: We removed the same part of the sentence.

8. A tuning range of up to 2 GHz is reported. Compared to other techniques this is a relatively small value, e.g., single sideband generation would easily allow tuning ranges between 10 to 20 GHz. Can this tuning range be increased and what would be needed for that?

In our work, we were limited by the laser self-injection locking range (2 GHz). It can be increased by increasing the backreflection from a Si₃N₄ circuit, e.g., by adding an on-chip loop mirror in the drop-port. [R. R. Galiev et al., "Mirror-assisted tuning of laser stabilization via self-injection locking to WGM microresonator," in Laser Congress 2021, paper JTU1A.40.]

Laser self-injection locking ranges around 10 GHz has been previously reported [<https://arxiv.org/abs/2103.07795>].

Another option we’ve already discussed in the FMCW demo section of the main manuscript is a tuning scheme where the laser diode current and the piezoactuator voltage are synchronously tuned

in a way that the laser is kept in an injection-locked state. In this case, a diode current tuning might not be precisely linear as self-injection locking implies and preserves the linearity of cavity resonance tuning by the piezoelectrical actuator. In such a feed-forward scheme, the laser tuning range would be limited by piezo actuators tuning range, potentially reaching 20 GHz for PZT actuators.

We would like to note that while SSB may indeed give fast and wideband frequency actuation, single-sideband generators are quite complex and bulky, as they require separate modulators and separate RF driving and VCO. In addition, additional phase noise is imparted on the shifted carrier.

9. “which has only been shown in Si₃N₄ low-confinement waveguides [3]”. A better result exists since ref. 3 has been published, see comment 4.

We thank the Referee for pointing out this reference.

Action taken: We added the cited work in the abstract and in the main text.

10. The main text reports a value for the FWHM linewidth for the 10-GHz FSR microring resonator while the 200-GHz FSR microring is used in the coherent ranging experiment. The FWHM should be reported for injection locking using the 200 GHz microring as that value is of interest for the coherent ranging experiment.

In the previous round of reviews, we added the intrinsic linewidth of all lasers to the main text. In this round of review, we also added integrated linewidth values for different integration times to the main manuscript. In combination with frequency noise plots, this constitutes a complete description of our laser noise performance, accepted in the community.

Action taken: For 190 GHz device, we added the intrinsic laser linewidth and integrated laser linewidth, including the integration time.

New sentence: “For the 190.7 GHz FSR Si₃N₄ device, the integrated linewidth is 43.27 kHz at 1 ms integration time, 73.7 kHz at 10 ms, and 81.7 kHz at 100 ms.”

11. Figure 6 in SI shows that the frequency noise below 1 kHz flattens, even for the free-running DFB laser. Why is this? If 1/f noise is dominant in this region, I would expect an increase of the frequency noise for smaller offset frequencies.

Additional technical (acoustic noise) is clearly present below 1 kHz offsets. The noise level is high (10^7 Hz²/Hz) for both reference Toptica laser and SIL laser (Figure 6 in the SI) and might dominate other noise sources. For SIL laser, we attribute it to the acoustic noise from xyz mechanical stages. No packaging of the DFB laser and Si₃N₄ has been done. 1/f slope is visible in SIL 2 GHz (blue trace) and PDH-locked Toptica (red trace) at offsets below 100 Hz. A longer measurement time is required for the heterodyne beatnote IQ datasets to investigate 1/f noise in more detail. We think that analysis of noise contribution factors is out of this paper's scope.

12. Why is there no information given on the accuracy of the measured single sided PSD?

We have checked the literature from comparison table II in the SI and could not find a case of a photonic integrated laser, where the linewidth was given with uncertainty (measured with the heterodyne method).

This is not done in the field; most groups (and also us) use the commercial calibration of high-end electrical spectrum analyzers that are ubiquitously used in the field. It is not common in the field, and the frequency noise measurement accuracy is given by showing the noise floor in the measurements, which for our measurement is: -110 dBm (Rohde&Schwarz FSW43, resolution bandwidth 10 kHz, analysis bandwidth 12.5 MHz).

That said, we have made a comparison that our frequency noise measurements are faithful. Specifically, we have used the commercial spectrum analyzer option for the heterodyne method and constructed an optical cross-correlation measurement. Importantly, our two techniques agree to better than 3 dB within the overlapping region.

13. “showcases the remarkable frequency agility of our system.” Indeed a good performance is demonstrated, however, similar if not better performance can be obtained using single sideband frequency generation followed by injection-locked amplification in a Bragg diode laser, which, in principle, could also be implemented on a single chip assembly. The authors need to put their work in perspective and discuss state-of-the-art.

It is true that fiber laser + SSB + circulator + injection locking setup has been shown with very good performance. We added the citation and discussion in the main text. However, as of today, such a system has not been demonstrated on a single chip. The RF chirp generator with 10s of dBm output power for modulation adds additional complexity. We believe that integrating SSB, circulator and high-frequency RF chirp generator on a single chip is a much harder task than the integration of our system, see for example [Xiang C et al., Science. 2021 Jul 2;373(6550):99-103], which only lacks piezoelectric actuation, which can be added in a back end of line process. We would like to point out that we had already added a very similar system demonstration by Wei et al. (10.1364/OE.23.004970) to the SI.

Action taken: We added a citation for the SSB, injection locking LiDAR demo and discussed it in the main text.

New sentence: “Most digital approaches towards photonic integrated FMCW LiDAR, which employ injection-locking of a high power laser diode to an electro-optically modulated sideband of a coherent laser can deliver excellent linearity and low noise, but today require bulk optical circulators and fiber laser oscillators for operation.”

14. “Together with more than 1 GHz tuning range, such a high actuation bandwidth exceeds the performance of common benchtop laser systems that rely on bulk piezo or electrooptic components,” This statement is not true, see, e.g., doi: 10.1109/JLT.2021.3050772

We thank the Referee for pointing out this reference. It is conceptually very similar (Injection locking of EOM sideband) to by Wei et al. (10.1364/OE.23.004970), which is Ref. 38 of the SI and part of the tabular comparison of laser systems.

Action taken: We added a citation to the mentioned paper to the main manuscript and removed “or electrooptic components” from the sentence.

15. Reference 44 is not a proper reference. This reference does not discuss the properties of the laser and is just a conference abstract which promises a lot without showing details. Better would be to use "P. FeneYROU, et al., Appl. Opt. 56(35) 9663-9675 (2017)" and "P. FeneYROU, et al., Appl. Opt. 56(35) 9676-9685 (2017)" of the same authors, which are regular articles providing more and detailed information.

We thank the Referee for pointing out a better reference than the chosen one.

Action taken: We changed reference 44 to P. FeneYROU, et al., Appl. Opt. 56(35) 9663-9675 (2017) and added the second reference.

16. The abbreviation FEM is not defined in the text.

We thank the Reviewer for pointing out the omission of an acronym.

Action taken: We define the finite element method (FEM) in the main text.

17. “We apply triangular chirp with 100 kHz frequency to ...”. First it should be “We apply a triangular chirp ...”. Further, this is two orders of magnitude smaller than what can be realized by the laser (as demonstrated earlier in the manuscript). Why is the full potential not used, i.e., why go through all the effort of showing a flat actuation response to high frequency if it is not used in the application?

10 MHz small-signal response only gives 100kHz tuning rate in linear chirp with 0.03% RMS nonlinearity over the full tuning range. We added the analysis of bandwidth required to achieve target linearity for a 100 kHz triangular chirp. Figure 2(a) shows the ideal triangular signal at 100 kHz frequency and simulated triangular signals with lowpass RC filtering applied for different filter cutoff frequencies. Figure 2(b) shows RMS nonlinearity vs RC-filter cutoff frequency for

Figure 2: (a) Ideal triangular signal (blue) with 100 kHz frequency, triangular signal with applied RC filter function with 1 MHz cutoff frequency (red) and 10 MHz (orange). Inset: zoom into the turning point. (b) RMS nonlinearity (normalized to the tuning range) vs RC-filter cutoff frequency for different analysis region: 100% of ramp (blue), central 90% of ramp (red), central 80% (orange).

different analysis interval: 100% of ramp-up (blue), central 90% of ramp (red), central 80% (orange).

Action taken: We added a new “Actuation bandwidth requirements for targeted chirp linearity” in the SI, explaining the bandwidth requirements for a 100 kHz chirp rate to achieve good chirp linearity. We added the correct article in the sentence.

18. “Obtained time-frequency plots are presented in Fig. 4(c) for the target and in Fig. 4.” Some editing issues from the previous version of the text?

We thank the Reviewer for pointing out this error.

Action taken: We fixed this editing issue.

19. “See attached to the SI documented interactive code for LiDAR data processing”. Improve grammar.

We thank the Reviewer for pointing out this bad sentence.

Action taken: We rewrite the sentence: “Interactive code for LiDAR data processing can be found in the data availability section.”

20. “No zero-padding was used in processing point cloud data”. Are figures 4d and 4(e-t) not different representations of the same data. Why is different data processing used for these figures (with and without zero-padding)

Figures 4(d) and 4(e-g) represent the same data. We did not use zero-padding in FFT in Fig.4(e-g) to make Figure 4(g) visually clear, showing clear discretization corresponding to the distance resolution of 12.5 cm.

21. “alleviates the need for external components ... single sideband modulators ...”. on-chip single sideband modulators have been available for a long time, see, e.g., doi: 10.1109/ICTON.2015.7193321

We added citation [Fandino, J., et al. Nature Photon 11, 124–129 (2017)] in the main text. However, FMCW LiDAR with SSB also requires an RF chirp synthesizer which is very expensive.

Action taken: We added this citation to the main manuscript

New sentence: “Most digital approaches towards photonic integrated FMCW LiDAR, which employ injection-locking of a high power laser diode to an electro-optically modulated sideband of a coherent laser can deliver excellent linearity and low noise [46,47], but today require bulk optical circulators and fiber laser oscillators for operation.”

22. In the conclusions the authors mention that the source is ideal for coherent ranging applications (among others). However, to operate in the linear regime, the output power is limited to 1.5 mW and external amplification was needed for the coherent ranging test.

In our LiDAR demo experiment, we amplified 1.5 mW output laser power to 10 mW with external EDFA. 10 m target distance was chosen due to the limited size of our laboratory. Even with 1.5 mW of output power, the target scene was properly reconstructed with lower beatnote SNR. We use the same collimator with 8 mm aperture for all experiments.

Power requirement for targets with low scattering is a common problem for all FMCW LiDAR implementations. A detailed discussion of power requirements vs range is presented in the cited literature [Behroozpour et al. (2017)].

To maintain a linear regime in the Si₃N₄ cavity on-chip splitter might be introduced. In such cases, e.g., 10% of DFB power is sent to the Si₃N₄ microresonator for self-injection locking, 90% of the power after splitter goes to the output port. In such a design, the laser always operates in a linear regime.

Such configuration is used in commercial lasers from OEwaves, where they use free-space optics (beamsplitter, prism coupler, lenses) in hybrid packaging of high-Q crystalline microresonators and diode lasers.

What would happen in the integrated laser was operated in the nonlinear regime? What changes and how would this affect the application?

Our device can be operated in a nonlinear regime at high DFB current and proper optical feedback phase [Voloshin, A.S. et al., Nat Comm. 12, 235 (2021)]. Laser self-injection locking of high power DFB to high Q Si₃N₄ with anomalous GVD allows generating soliton microcombs. 10 GHz soliton microcomb data has been removed from this manuscript after the Reviewer's suggestion in the second round of reviews.

We have also investigated comb tuning with an integrated AlN actuator for a 190 GHz device. During soliton microcomb generation, the laser remains narrow linewidth and highly coherent and can be modulated over a full 1 GHz range while retaining a single soliton state. The modulation instability region of Kerr comb generation must be avoided for coherent FMCW operation.

Do the authors foresee that this technology would allow for coherent ranging without additional amplification, i.e., can the self-injection locked laser provide sufficient power for the application by itself?

Yes, we foresee our technology to be a relevant candidate for short-range coherent ranging without external amplification. To realize such power levels, we would need to further improve the laser to chip coupling loss and increase the optical feedback strength, for example, by adding a drop-port loop mirror. The technology can also be used for long-range fiber sensing.

Based on our discussions with several companies in the LiDAR market, output optical powers in mW range at telecom wavelengths are sufficient for their needs. They also have reliable solutions for post-amplification (e.g. SOAs), which can also be integrated with our laser system in the same manner as the laser source.

In conclusion, I cannot recommend publication of the manuscript in its current form.

We regret the decision of the Referee and hope that our revised manuscript and the review reply address all concerns of the Referee.

REVIEWERS' COMMENTS

Reviewer #2 (Remarks to the Author):

All points have been sufficiently addressed by the authors. The manuscript is significantly improved over all previous versions. The manuscript accurately states the novelty of the work, and more clearly explains the methodology as well as accurate referencing to related works. I can now recommend this paper for publication in Nature Communications.

REVIEWER COMMENTS

We are grateful that Referees have again seen our manuscript. We thank Reviewer #2 for his/her appreciation of the revised manuscript and positive comments on novelty and methodology.

Sincerely Yours,

Tobias J. Kippenberg

Reviewer #2 (Remarks to the Author):

All points have been sufficiently addressed by the authors. The manuscript is significantly improved over all previous versions. The manuscript accurately states the novelty of the work, and more clearly explains the methodology as well as accurate referencing to related works. I can now recommend this paper for publication in Nature Communications.